# Octupole moment driven free charge generation in partially chlorinated subphthalocyanine for planar heterojunction organic photodetectors

Aniket Rana[1,8], Song Yi Park[2,6,8], Chiara Labanti [2,8], Feifei Fang[3,8], Sungyoung Yun[3], Yifan Dong [1,7], Emily J. Yang [2], Davide Nodari [1], Nicola Gasparini [1], Jeong–Il Park[3], Jisoo Shin[3], Daiki Minami [4], Kyung-Bae Park [3] ✉, Ji-Seon Kim [2] ✉ & James R. Durrant [1,5] ✉

In this study, high-performance organic photodetectors are presented which utilize a pristine chlorinated subphthalocyanine photoactive layer. Optical and optoelectronic analyses indicate that the device photocurrent is primarily generated through direct charge generation within the chlorinated sub-phthalocyanine layer, rather than exciton separation at layer interfaces. Molecular modelling suggests that this direct charge generation is facilitated by chlorinated subphthalocyanine high octupole moment ($-80$ DÅ$^2$), which generates a 200 meV shift in molecular energetics. Increasing the thickness of chlorinated subphthalocyanine leads to faster response time, correlated with a decrease in trap density. Notably, photodetectors with a 50 nm thick chlorinated subphthalocyanine photoactive layer exhibit detectivities approaching $10^{13}$ Jones, with a dark current below $10^{-7}$ A cm$^{-2}$ up to $-5$ V. Based on these findings, we conclude that high octupole moment molecular semiconductors are promising materials for high-performance organic photodetectors employing single-component photoactive layer.

Organic photodetectors (OPDs) are attracting extensive attention due to the strong light-absorbing ability of organic semiconductors, their adjustable band gaps by molecular design, and facile processability by vacuum or solution techniques. Usually a bulk heterojunction (BHJ) structure is used to enable the efficient separation of photogenerated excitons into the charges required for OPD function. An alternative approach is to fabricate planar heterojunction (PHJ) photodiodes based on the sequential deposition of separate layers of electron donor and acceptor materials. The PHJ structure has been reported to be more morphologically stable compared to BHJs due to the absence of often poorly controlled phase separation[1]. PHJ structures have also been suggested to facilitate lower energetic disorder due to the separate optimisation of each layer, better-defined interfaces, and purer domains, reduced bimolecular recombination losses, and

[1]Department of Chemistry and Centre for Processable Electronics, Imperial College London, London W12 0BZ, UK. [2]Department of Physics and Centre for Processable Electronics, Imperial College London, London SW7 2AZ, UK. [3]Organic Materials Lab, Samsung Advanced Institute of Technology, Samsung Electronics Co. Ltd., Samsung-ro, Yeongtong-gu, Suwon-si, Gyeonggi-do 16678, Republic of Korea. [4]Innovation Center, Samsung Electronics Co. Ltd., 1 Samsungjeonja-ro, Hwaseong-si, Gyeonggi-do 18448, Republic of Korea. [5]SPECIFIC, Faculty of Science and Engineering, Swansea University, Swansea SA1 8EN, UK. [6]Present address: Department of Physics, Pukyong National University, Busan 48513, Republic of Korea. [7]Present address: National Renewable Energy Laboratory, 15013 Denver W Pkwy, Golden, CO 80401, USA. [8]These authors contributed equally: Aniket Rana, Song Yi Park, Chiara Labanti, Feifei Fang. ✉e-mail: myshkin.park@samsung.com; ji-seon.kim@imperial.ac.uk; j.durrant@imperial.ac.uk

enhanced charge transport across the device[2,3]. However, the performance of PHJ OPDs is typically limited by exciton recombination losses during their diffusion to the donor/acceptor interface, resulting in relatively poor OPD quantum efficiencies compared to BHJ devices. Usually, $C_{60}$ is the most widely employed electron acceptor in both PHJ and BHJ OPDs. However, several studies have highlighted the potential of subphthalocyanines (SubPc) derivatives as alternative acceptors, with the particular benefit of stronger and narrower bandwidth light absorption[4,5]. Moreover, SubPc halogenation can tune the highest occupied molecular orbital (HOMO) and lowest unoccupied molecular orbital (LUMO) levels for device optimisation and potentially suppress recombination processes[6]. Some recent reports have demonstrated the use of PHJs based on SubPc in organic electronic applications, including organic solar cells (OSCs) with high open circuit voltage[7,8] and OPDs for full-spectrum detection[9]. However, the low mobility and short exciton diffusion lengths of typical amorphous SubPc derivatives have limited the optimum layer thickness in PHJs, which thus often show lower performances compared to BHJs[10,11]. In this context, chlorinated subphthalocyanine ($Cl_6$-SubPc) delivers promising characteristics among other SubPc derivatives, owing to its strong intermolecular interactions and good electron transport[12]. Moreover, the strong and sharp absorption of $Cl_6$-SubPc allows efficient exciton generation in <20 nm thick layers, enhancing the potential for efficient PHJ OPD performance. In addition to, charge pair dissociation at the interface with matched donor materials[13], SubPc derivatives have also shown efficient long-range Förster resonant energy transfer to selected donor materials, resulting in efficient OSCs[14]. More recently, OSCs based upon a neat SubPc photoactive layer have been reported, albeit with only modest device efficiencies[15]. In this study, we focus on $Cl_6$-SubPc's potential for efficient OPDs based on direct charge generation within a neat $Cl_6$-SubPc photoactive layer.

Herein, we study evaporated $Cl_6$-SubPc based planar OPDs, fabricated as heterojunctions with a thin 2-((8-methyl-8H-thieno[2,3-b]indol-2-yl)methylene)-1H-cyclopenta[b]naphthalene-1,3(2H)-dione) (MPTA) layer[16]. OPD devices are studied as a function of $Cl_6$-SubPc thickness, focusing particular on photocurrent generation from photoexcitation of $Cl_6$-SubPc acceptor top layer, with the MPTA primarily functioning as a hole collection layer. It is observed that these photodiodes with a thick (50 nm) $Cl_6$-SubPc layer show promising OPD performance with external quantum efficiencies of up to 35%, specific detectivities close to $10^{13}$ Jones, 0.15 A $W^{-1}$ of responsivity and 5 μs of response time at −1 V as well as high negative bias breakdown tolerance up to −5 V. Transient absorption spectroscopy along with bias dependent external quantum efficiency (EQE) and photoluminescence (PL) data indicate that this high-performance results from efficient free charge carrier generation within the $Cl_6$-SubPc layer, highlighting the potential of $Cl_6$-SubPc as light absorber layer in efficient PHJ OPDs. Density functional theory (DFT) calculations indicate $Cl_6$-SubPc exhibits an unusually high octupolar moment. Calculated energetic offsets within the $Cl_6$-SubPc layer resulting from this high octupole moment are suggested to be the origin of the observed direct charge generation. The potential importance of molecular octupole moments determining a short-range electrostatic potential in optimising OPD performance has not been reported previously. Overall, the $Cl_6$-SubPc's efficient direct charge generation property is highly significant for OPD applications, as it can be utilized to create efficient single-component as well as multi-layer devices.

## Results
### Optical and optoelectronic analyses
Figure 1a, b shows the molecular structures of MPTA (donor) and $Cl_6$-SubPc (acceptor) as well as the OPD device structure employed. We start by considering the steady-state absorption and emission properties of neat donor and acceptor films as well as PHJ films with varying acceptor layer thicknesses. Absorption spectra maxima for MPTA and

$Cl_6$-SubPc are observed at 494 nm and 586 nm respectively, with PHJ absorption increasingly dominated by $Cl_6$-SubPc as the thickness of this acceptor layer is increased (Fig. 1c). Photoluminescence spectra exhibit peaks at 650 nm for neat $Cl_6$-SubPc and 750 nm for MPTA films (Fig. 1d), from which it is apparent that MPTA exhibits a red-shifted emission relative to $Cl_6$-SubPc, previously attributed to formation of emissive donor/acceptor (D/A) charge transfer (CT) states[16]. It is striking from Fig. 1d that the PL spectra of the PHJ films do not show reduced PL relative to the neat films, and indeed for the thinner bilayers show enhanced photoluminescence. This contrasts with data typically observed for organic BHJs or PHJs, where D/A interface formation is normally correlated with strong PL quenching, assigned to exciton separation at the interfaces. The absence of PL quenching for the PHJs studied herein strongly indicates that there is negligible exciton separation at the D/A interface. This is consistent with reported exciton diffusion lengths of SubPc's of 7 nm to 28 nm, shorter than the layer thickness[11,17,18], such that photogenerated excitons are unable to reach the D/A interface. Consistent with this conclusion, PL from $Cl_6$-SubPc is quenched in case of PHJ 10/10 nm, as 10 nm of the $Cl_6$-SubPc layer is shorter than its reported exciton diffusion length. This observation therefore indicates that any photocurrent generation in these devices does not primarily result from exciton separation at the MPTA/$Cl_6$-SubPc interface. The HOMO energy levels of neat and bilayer films measured by ambient photoemission spectroscopy[19] are summarised in Fig. 1e, f. Along with confirming the expected energetic offset between MPTA and $Cl_6$-SubPc, these data also indicate that the HOMO level of $Cl_6$-SubPc is essentially independent of layer thickness. The thickest layer (60 nm) shows a slightly deeper HOMO level, which can be attributed to improved $Cl_6$-SubPc molecular alignment[20].

Current density–voltage (J–V) measurements were performed under 1 sun and dark conditions to understand the effect of $Cl_6$-SubPc thickness on the OPD photocurrent, as well as measured trap density under dark conditions shown in Fig. 2a–c, respectively. It is evident that under light illumination a higher photocurrent is obtained as the thickness of $Cl_6$-SubPc increases, which can be attributed to enhanced light absorption in the thicker $Cl_6$-SubPc devices. The abrupt current increase for the PHJ 10/10 nm device after −4 V is due to diode breakdown. EQE spectra as a function of reverse bias are shown in Fig. 2d–f. As the $Cl_6$-SubPc thickness is increased, there is an increased dominance of photocurrent generation from $Cl_6$-SubPc light absorption. For all devices, the EQE is observed to increase with reverse bias. It is apparent that the thickest device, with a 50 nm $Cl_6$-SubPc layer, exhibits the highest EQE, peaking at >40% at −5 V, consistent with this device yielding the highest photocurrent under white light irradiation. The thicker layer will result in significantly higher light absorption. Whilst interference effects may also result in enhanced light absorption, device reflectance data (Supplementary Fig. 1) indicate such effects will only be minor and insufficient to explain the observed trend. We note that detailed analyses of such interference effects are beyond the scope of this paper. The observation of the highest EQEs and photocurrent densities for devices with 50 nm of $Cl_6$-SubPc is particularly striking, as for most PHJ OPDs, photocurrent is suppressed for thicker layers due to the layer thickness becoming greater than their exciton diffusion length. As aforementioned, PL data of PHJ films indicate negligible $Cl_6$-SubPc exciton quenching at the MPTA/$Cl_6$-SubPc interface (Fig. 1d), consistent with this 50 nm film thickness being much larger than $Cl_6$-SubPc exciton diffusion length[21]. These EQE and PL data therefore strongly indicate that the photocurrent generation does not originate from exciton separation at the MPTA/$Cl_6$-SubPc interface but rather from direct photogeneration charges within the bulk of $Cl_6$-SubPc layer, as we discuss further below.

It is apparent from J-V characteristics of OPDs device data that the thickest PHJ 10/50 nm device shows the lowest dark current (Fig. 2b). This is correlated with a lower trap density determined from impedance analyses of device frequency-dependent capacitance response

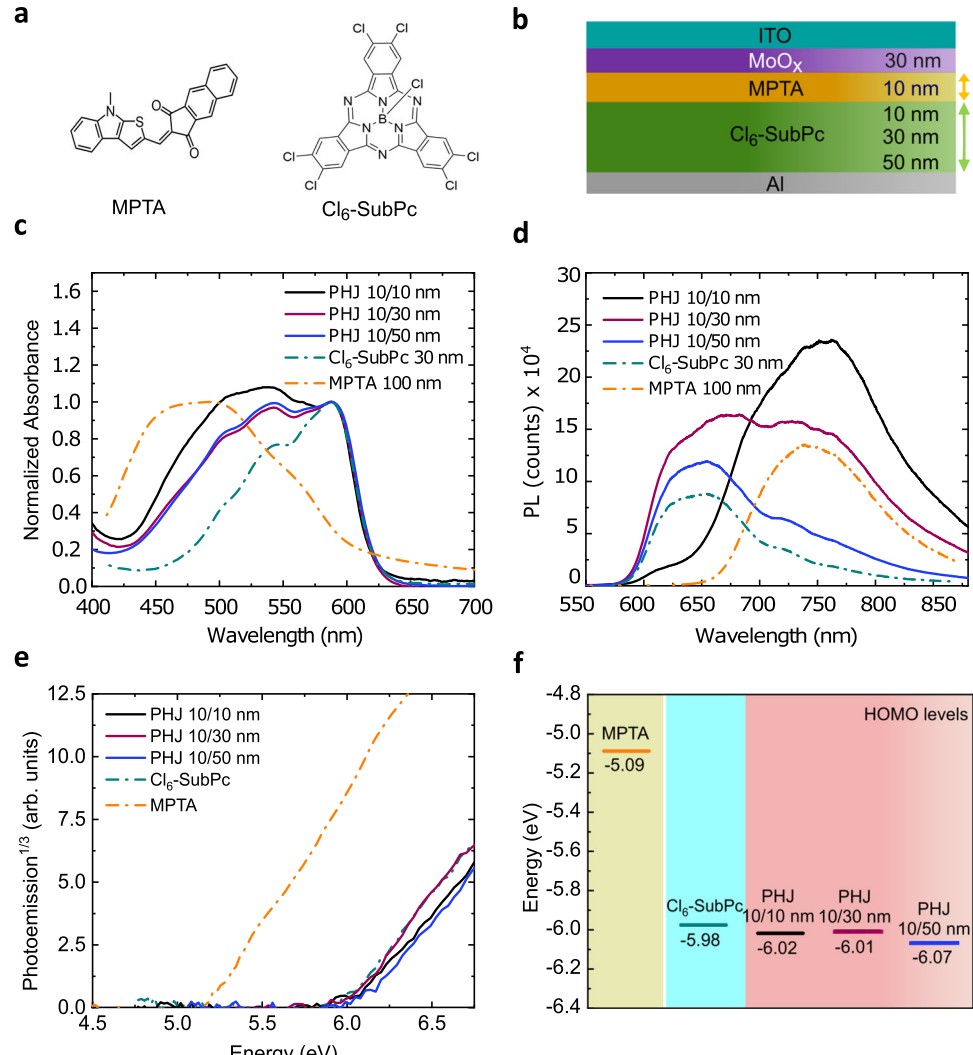

**Fig. 1 | Thickness-dependent properties of neat MPTA and MPTA/Cl₆-SubPc PHJ films. a** Molecular structure of MPTA (donor) and Cl₆-SubPc (acceptor). **b** Schematic of PHJ device architecture with varying acceptor thickness. **c** Normalized UV-Vis absorption spectra and (**d**) Absolute photoluminescence spectra (measured with 514 nm excitation and normalised to matched densities of absorbed photons). **e** Ambient photoemission spectra and (**f**) HOMO levels for neat MPTA and Cl₆-SubPc as well as PHJ films with different thicknesses of Cl₆-SubPc.

under dark (Fig. 2c, see also Supplementary Fig. 2a–c)[22]. This technique provides the detection of trap density from the capacitive response[23,24] and has been successfully applied in organic photodiodes and OSCs[24,25]. Trap responses above 0.65 eV are likely to be dominated by the resistance of device layers and therefore do not correspond to deep states[24]. The σ (disorder parameter)[26] is evaluated from reconstructed trap density after fitting with a Gaussian function[27], yielding values of 52 meV, 42 meV and 30 meV for the 10/10 nm, 10/30 nm and 10/50 nm PHJ devices respectively, with most of the traps centred around 0.48 eV. From these analyses, it is apparent that the thinnest devices show the highest trap density, correlated with its highest dark current. We have also qualitatively evaluated the density of trap states in these PHJ devices from ambient photoemission spectra, by integrating tail area below the band edge (see Supplementary Fig. 3)[19,28]. From these photoemission data, we also observe the highest sub-bandgap tail states in PHJ 10/10 nm device whereas PHJ 10/50 nm device shows the least sub-bandgap tail states, in agreement with the capacitance analysis. Interestingly, the thinnest device also shows the highest interfacial trap density per unit area (see Supplementary Fig. 4).

We also plotted the dark current versus electric field (accounting for the difference in field with organic film thickness), as shown in Supplementary Fig. 5; in this plot, the dark current is still highest (by over an order of magnitude difference at low field) for the thinnest device. All of these analyses are indicative of a lower trap density in the thickest device, which is likely to contribute to lower dark current in this device. A previous study has shown that trap states can arise from structural and chemical defects within material[29]. The suppression of trap density for the thicker films is consistent with a previous report that thicker SubPc layers are structurally more ordered[30], less affected by pinholes and ITO spikes, therefore, exhibit reduced shunt paths in device. Also, it has been reported that thicker and more crystalline organic films suffer less destruction at their surface during aluminium deposition[31].

**Origin of intrinsic free charge generation in Cl₆-SubPc**

In order to further investigate the mechanism of charge photogeneration in these OPDs, bias-dependent PL measurements were undertaken on PHJ devices, as shown in Fig. 3a–c (see Supplementary Fig. 6a, b for a comparison of the PL quenching efficiency and EQE as a

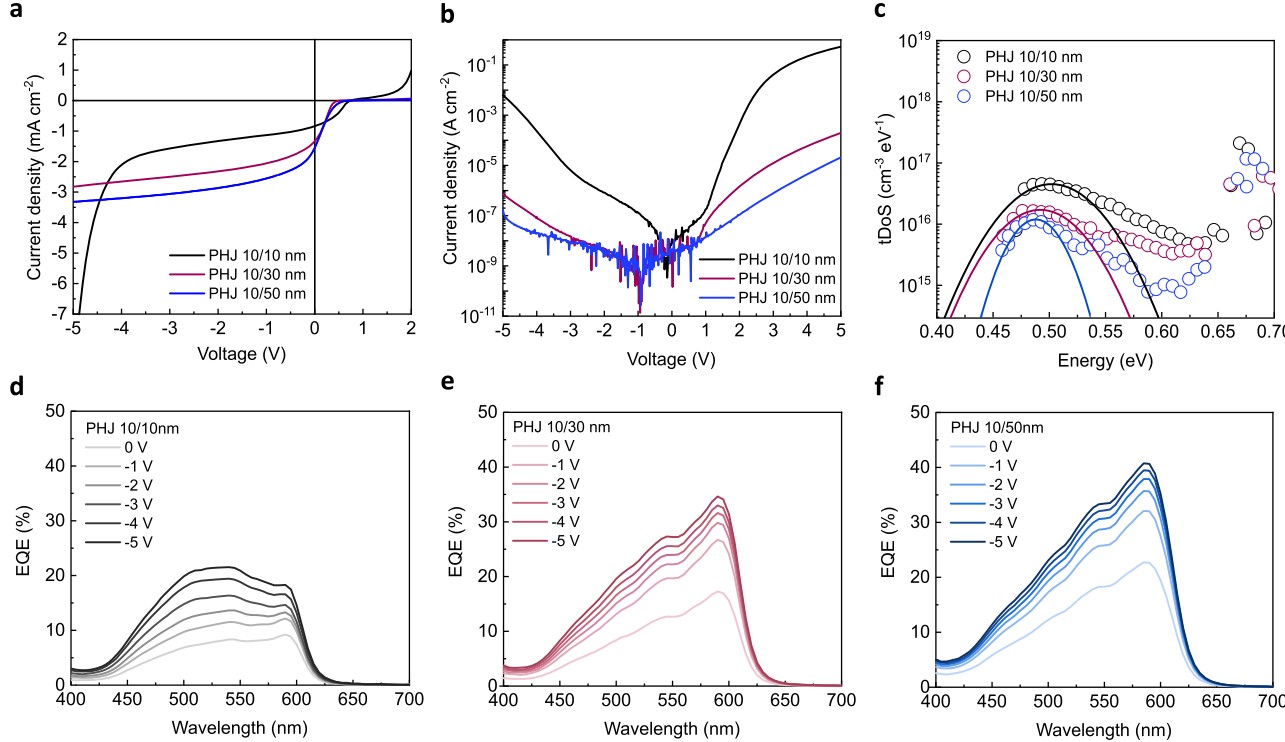

**Fig. 2 | Thickness-dependent PHJ OPD devices characterization. a** Current density-voltage characteristics of PHJ MPTA/Cl$_6$-SubPc photodiodes with different thicknesses of acceptor under 1 sun and (**b**) under dark conditions. **c** Trap density distribution per unit volume determined from capacitance-frequency measurements. External quantum efficiency spectra as a function of applied electric bias for (**d**) PHJ 10/10 nm (**e**) PHJ 10/30 nm and (**f**) PHJ 10/50 nm device.

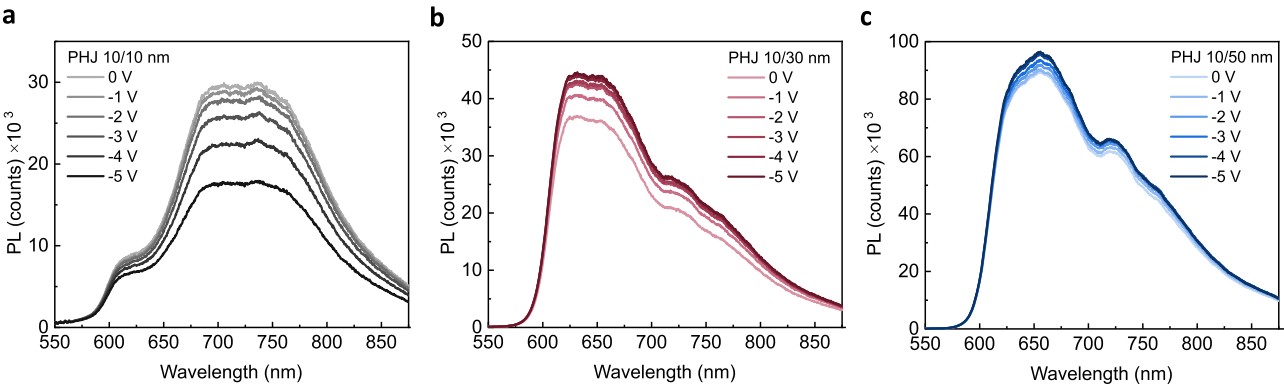

**Fig. 3 | Thickness-dependent photoluminescence characterization of PHJ OPD devices.** Bias-dependent photoluminescence spectra of different thickness of bilayer devices, following excitation at 514 nm (**a**) PHJ 10/10 nm, (**b**) PHJ 10/30 nm and (**c**) PHJ 10/50 nm MPTA/Cl$_6$-SubPc photodiodes, respectively.

function of electric field, respectively). For the PHJ 10/10 nm device, where the PL around 650 nm is dominated by MPTA emission, this PL is observed to be strongly quenched by negative applied voltages. This is consistent with our previous reports of the field-dependent separation of emissive CT states formed between MPTA and the acceptor (C$_{60}$)[16]. We note the PL intensities of these devices are likely to be impacted by interference (and possible quenching) effects resulting from the presence of the aluminium (Al) top contact, as such we focus here only on the bias dependence of device PL. This is also consistent with the bias dependence of the EQE being more pronounced for MPTA rather than Cl$_6$-SubPc optical excitation in these devices (see Fig. 2d), indicative of charge photogeneration in PHJ 10/10 nm device primarily deriving from the field-dependent separation of MPTA/Cl$_6$-SubPc CT states. In contrast, for the devices with thicker (30 nm and 50 nm) Cl$_6$-SubPc layers, where the PL around 650 nm (and EQE around 590 nm) are dominated by Cl$_6$-SubPc, negligible PL quenching is observed with

negative bias (indeed a small PL increase is observed), indicating that Cl$_6$-SubPc excitons are not significantly impacted by the applied field. This suggests that any charge photogeneration in Cl$_6$-SubPc is likely to be an intrinsic property of the Cl$_6$-SubPc, rather than a result of applied electric fields.

We turn now to ultrafast transient absorption spectroscopy characterisation to further investigate the possibility of direct charge photogeneration in Cl$_6$-SubPc films. Figure 4a shows the transient absorption spectra for a pristine Cl$_6$-SubPc thin film following excitation at 610 nm. At early times, the spectra are dominated by a sharp negative ground state bleach/stimulated emission feature, indicative of the dominance of Cl$_6$-SubPc singlet excitons. At longer time delays this feature reduces in amplitude and blueshifts to 585 nm, with the resultant spectrum becoming more structured and derivative like. Such derivative spectra are typical of electro-absorption spectra for charge pairs in organic semiconductors[32]. A genetic algorithm-based global

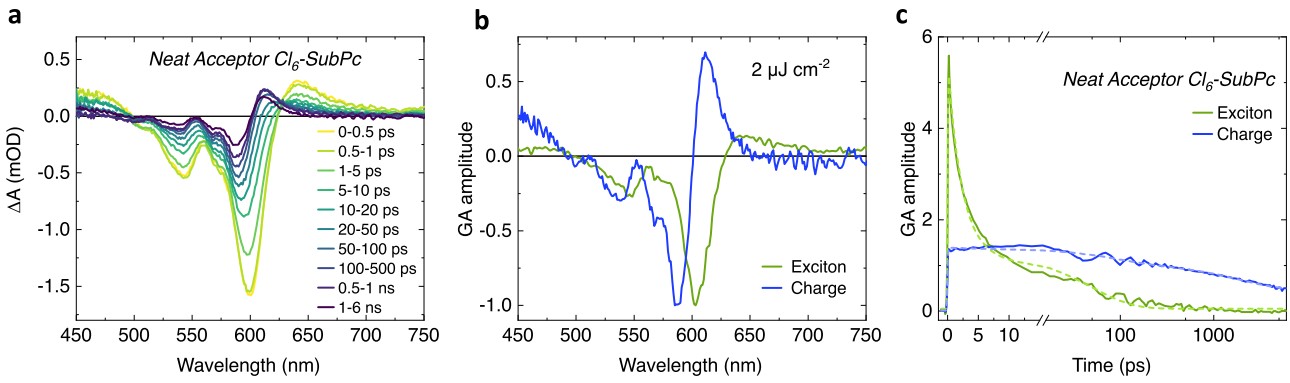

**Fig. 4 | Transient absorption (TA) characterisation. a** $Cl_6$-SubPc TA spectra as a function of time delay for pristine acceptor $Cl_6$-SubPc film following pulsed $2\,\mu J\,cm^{-2}$ excitation at 610 nm. **b** Exciton and charge species spectra and (**c**) TA kinetics determined from two component global analysis.

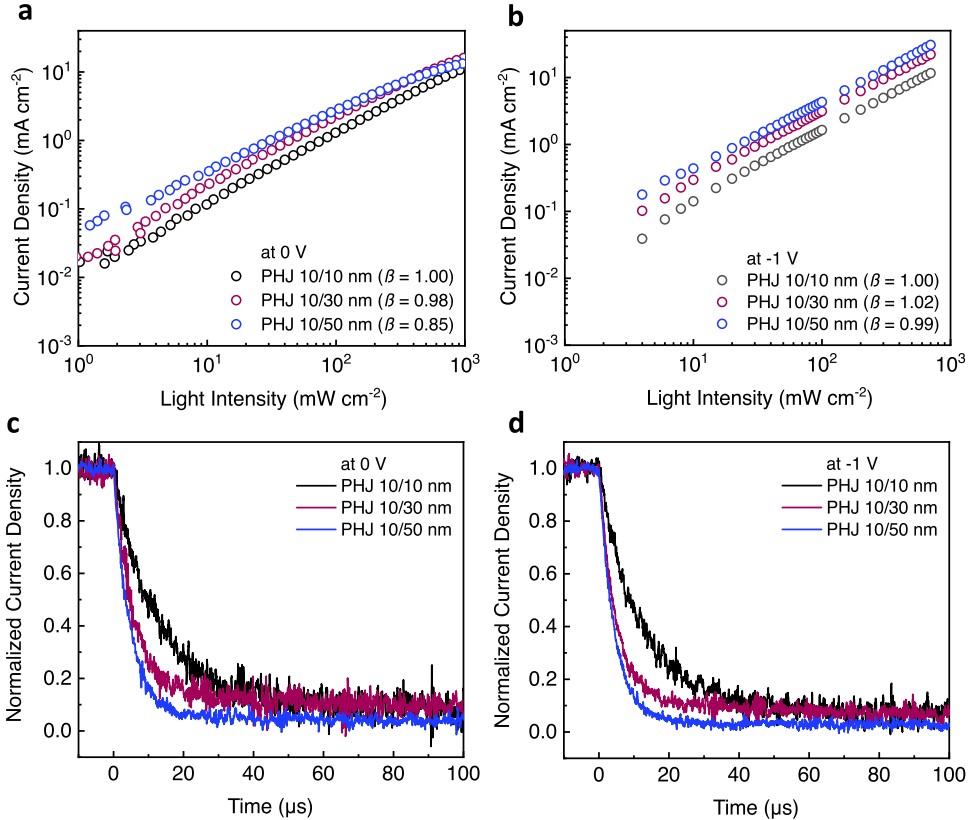

**Fig. 5 | Bias-dependent PHJ OPDs device performance.** Photocurrent densities versus light intensity under (**a**) Short circuit (0 V) and (**b**) −1 V bias conditions. Normalized transient photo current under (**c**) 0 V and (**d**) −1 V by illuminating the devices by a green laser diode (523 nm) with estimated light intensity of $4.22\,mW\,cm^{-2}$.

analysis was carried out to analysis these data further, which extracted two components whose spectra are illustrated in Fig. 4b and assigned to photogenerated excitons (green spectrum) and charges (blue spectrum). The population dynamics of these excitons and charges are plotted in Fig. 4c. Analogous analyses for different excitation densities (4 and $8\,\mu J\,cm^{-2}$) are shown in Supplementary Fig. 7a–d. From these data, it is apparent that $Cl_6$-SubPc excitons lifetime exhibit excitation density dependence and fast (<10 ps half time) decays, assigned to exciton−exciton annihilation due to the high densities of excitons generated in such transient measurements[33,34]. More strikingly, the structured spectral feature assigned to photogenerated charges appears within our instrument time response (circa 200 fs), indicating ultrafast (<200 fs) photogeneration of charges, rather than charge generation resulting from exciton decay on longer timescales.

The ultrafast appearance of this signal also allows us to rule out assignment of this signal to triplet excitons. There have been several recent reports of analogous ultrafast charge generation in organic thin films, including $\alpha$-6T[35,36] and Y6 thin films[37]. Further support for ultrafast charge photogeneration within $Cl_6$-SubPc comes from analogous transient absorption data for MPTA/$Cl_6$-SubPc bilayers (Supplementary Fig. 8a–c), which, apart from small differences for the thinnest 10/10 nm bilayer, exhibit transient absorption data almost indistinguishable from neat $Cl_6$-SubPc. The charge signal decays faster at higher laser intensities (Supplementary Fig. 7a–d, indicative of the bimolecular recombination of separated charges, although the formation of some bound charge pairs cannot be excluded. This observation of direct photogeneration of charges in a $Cl_6$-SubPc thin film is consistent with the conclusions of our bias-dependent bilayer PL data

and OPDs EQE above, which indicate that photocurrent generation following $Cl_6$-SubPc light absorption does not result from exciton quenching at the MPTA/$Cl_6$-SubPc interface but is rather an intrinsic property of the $Cl_6$-SubPc thin film.

We turn now to more in-depth analysis of the optoelectronic response of the MPTA/$Cl_6$-SubPc OPD devices. Figure 5a, b shows light intensity-dependent photocurrent densities and their linearity for two different bias conditions respectively, with derivative data shown in Supplementary Fig. 9. Under short-circuit conditions, the slope ($\beta$) of current density vs. light intensity (in log-log scale) is near unity for the thinnest (10/10 nm) device but decreases to 0.98 for PHJ 10/30 nm and 0.85 for PHJ 10/50 nm. This reduction in slope for thicker $Cl_6$-SubPc layers is indicative of increased bimolecular recombination losses within the $Cl_6$-SubPc layer at higher light intensities (see also Supplementary Fig. 9a, b for derivative data). This observation is consistent with free charge carrier generation within the bulk of $Cl_6$-SubPc layer competing with charge transport and extraction. However, under the reverse bias conditions typically employed in OPD devices under operation (e.g. −1 V as shown in Fig. 5b, and Supplementary Fig. 9b, all three OPDs show excellent photocurrent linearity, with near unity slopes, indicating efficient charge collection and negligible bimolecular recombination losses. Therefore, it can be concluded that even modest applied fields are sufficient to efficiently extract photogenerated charges from the $Cl_6$-SubPc layer, suggesting effective, ambipolar charge transport in this layer.

Transient photocurrent measurements were employed to assay the charge transport and extraction properties of OPDs as a function of applied bias and $Cl_6$-SubPc layer thickness. Photocurrent transients were observed to be relatively insensitive to applied bias (Fig. 5c, d). However, the thinnest PHJ 10/10 nm devices show the slowest transport time ($\tau$) of 11.3 μs, whereas PHJ 10/30 nm and PHJ 10/50 nm devices show faster $\tau$ around 5 μs (a similar trend was also observed at low white light intensity, see Supplementary Fig. 10).

As the thickness increases, the illuminated capacitance response drop shifts to higher frequencies, indicating that the thick devices have a faster response, consistent with their transient photoresponse, as shown in Supplementary Fig. 11. It is thus apparent that the response time of these devices is not limited by charge transport kinetics within the $Cl_6$-SubPc layer. The slower response time of the thinnest device, more dominated by MPTA light absorption, could result from slower charge transport kinetics within the MPTA. However, it appears more likely that it is correlated with the higher trap densities observed for the thinnest devices from our impedance and photoemission spectroscopy analyses, as discussed above.

## Molecular origin of direct charge photogeneration

Dielectric analyses of single layer $Cl_6$-SubPc films without MPTA (see Supplementary Fig. 12) indicate a $Cl_6$-SubPc dielectric constant of circa 5 and also independent of frequency over the range measured (up to 200 kHz). This dielectric constant is consistent with those reported for films of analogous molecules (5.8 for SubNc and 4.3 for SubPc)[38,39], but higher than most other organic films (typically between 2 and 4)[40]. SubPc derivatives have also been reported to have high optical constants, attributed to their cone-shaped molecular structure and solid-state packing[41,42]. This higher dielectric constant can be expected to reduce the exciton binding energy and thus favour the generation of free charges in the neat $Cl_6$-SubPc films. Nevertheless, when compared to inorganic materials, the dielectric constant of $Cl_6$-SubPc is still low[43], and it appears unlikely that the remarkably efficient direct charge generation observed for $Cl_6$-SubPc herein can be attributed to its higher dielectric constant alone. In order to explore further the charge generation properties of $Cl_6$-SubPc, we turn now to DFT calculations of isolated $Cl_6$-SubPc molecules. We and others have previously highlighted the potential of molecular quadrupole moments in driving charge generation in single component devices[37,44]. However, as shown

in Supplementary Fig. 13, $Cl_6$-SubPc exhibits a negligible (>10 times smaller) quadrupole moment, compared to common acceptor molecules, ruling this out as a potential origin of the observed efficient direct charge generation[44]. Instead, our calculations indicate (Fig. 6a) that $Cl_6$-SubPc possesses an anomalously high octupole moment (in the π–π stacking direction) of around −80 DÅ², which is significantly higher than MPTA as well as other widely studied non-fullerene acceptor molecules. This conclusion is consistent with previous reports of high octupole moments for SubPc[45].

Molecular interactions resulting from octupole moments can be expected to be relatively short-range as it is third-order term in the multipole expansion of molecular moment[46]. Therefore, cluster DFT calculations were employed to investigate whether $Cl_6$-SubPc's octupole moment is sufficient to impact the film energetics. An amorphous morphology was first generated through the simulation of 1000 $Cl_6$-SubPc molecules as shown in Fig. 6b. The influence of the octupole partial charges from surrounding molecules on the HOMO energy of each $Cl_6$-SubPc was assessed by analysing 200 randomly selected clusters of $Cl_6$-SubPc, as depicted in Fig. 6c. Calculations were undertaken in the absence and presence of these partial charges, as shown in Fig. 6d. It is apparent that the inclusion of the partial charges results a 200 meV shift in the distribution HOMO level maxima from −5.70 eV to −5.90 eV, with the latter value aligning closely with the energetics obtained from photoemission measurements (Fig. 1f). A similar shift was also observed in the LUMO energy, with the resultant bandgap thus being independent of partial charges (Supplementary Fig. 14). These calculations thus indicate that the electrostatic potential induced by $Cl_6$-SubPc's high octupole moment is sufficient to drive an average 200 meV shift in the energy levels of $Cl_6$-SubPc molecules embedded within $Cl_6$-SubPc clusters. It is also striking that the inclusion of partial charges increases the FWHM of the calculated HOMO energies from 150 meV to 440 meV, indicating that the energetic shifts caused by the partial charges are strongly dependent on the local molecular packing. This broad distribution of energy levels induced by the partial charges results in large (100's meV) energetic offsets between different regions (e.g. clusters) within the film, dependent on local molecular packing, which can be expected to aid exciton separation into separated charges (we note that energetic offsets of similar magnitude have been shown to be sufficient to drive exciton separation in bulk-heterojunction organic solar cell[47]). This was further explored by calculating the energetic offset between bulk and grain boundaries within a crystalline $Cl_6$-SubPc film, as detailed in Supplementary Figs. 15a–d. These calculations indicate an average HOMO level energetic offset of 45 meV between the bulk and grain boundaries in the presence of partial charges (with no offset being observed in the absence of partial charges). Such a shift is analogous to, but smaller than, molecular quadrupole-induced shifts between bulk and surface energetics we have reported previously for Y6 and α-6T linked to direct charge photogeneration in this systems[36,44]. For $Cl_6$-SubPc, it appears that the largest partial charge-induced energetic shifts result from differences in local molecular packing rather than between bulk and grain boundaries, consistent with expected shorter range of octupole interactions. These results also confirm that any charge photogeneration in $Cl_6$-SubPc is likely to be an intrinsic property of the $Cl_6$-SubPc (i.e. due to its high octupole moment) rather than a result of applied electric fields, consistent with no PL quenching under applied electric fields. We note that the octupole-induced energetic shifts may also impact upon charge transport, dependent upon film nanomorphology, a topic beyond the scope of this study. It is likely that the direct (instrument response limited) charge generation indicated by our transient absorption data results from photoexcitation occurring in regions where the local molecule packing results in large octupole-induced energetic offsets. In summary, our DFT calculations indicate that local energetic offsets induced by octupole interactions between $Cl_6$-SubPc molecules are likely to be sufficient to be the primary origin

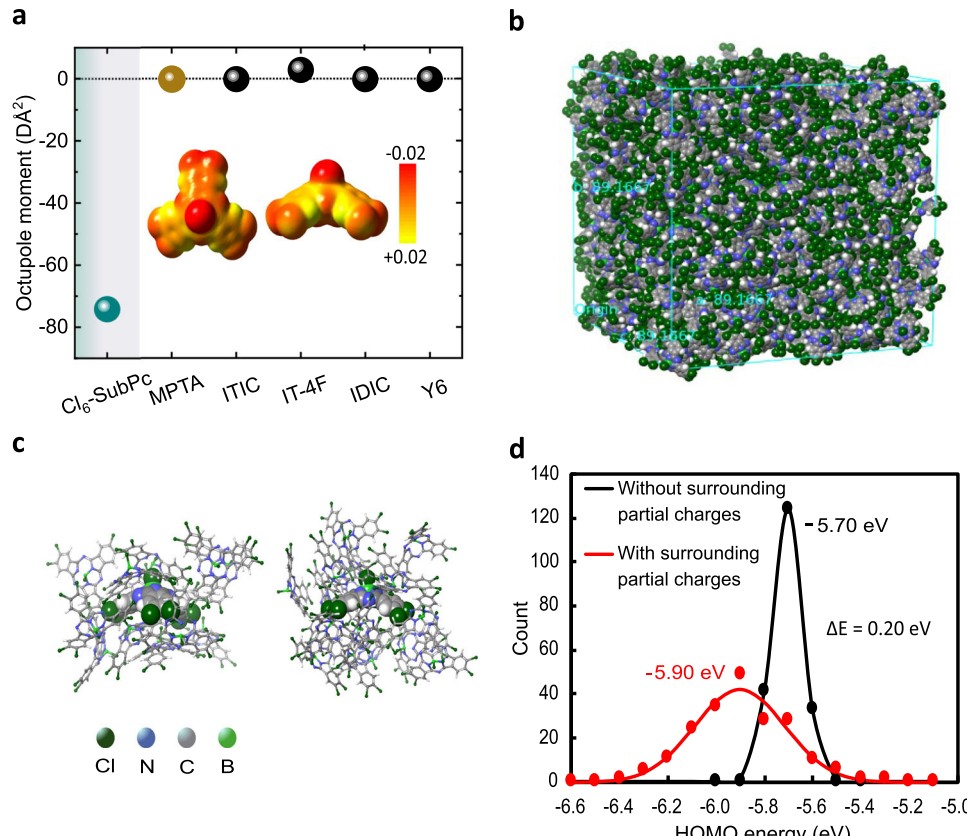

**Fig. 6 | DFT simulations for single molecule/cluster structures. a** Single molecule DFT calculations of the octupole moment (in the π-π stacking direction) of $Cl_6$-SubPc, the donor molecule (MPTA) and other common non-fullerene acceptors. The inset shows top and side views of the electrostatic potential distribution in $Cl_6$-SubPc which generate its high octupole moment. **b** Illustration of the amorphous structure determined from molecular dynamics (MD) simulations for 1000 molecules of $Cl_6$-SubPc. **c** Two of the 200 different clusters of $Cl_6$-SubPc's were randomly sampled from the structure in (**b**) for calculation of molecular energetics. Green, blue, grey, and light green balls indicate chlorine (Cl), nitrogen (N), carbon (C), and boron (B) atoms of $Cl_6$-SubPc respectively. **d** HOMO energy level distributions of the central $Cl_6$-SubPc's in each cluster calculated without/with inclusion of the partial charges associated with $Cl_6$-SubPc's high octupole moment.

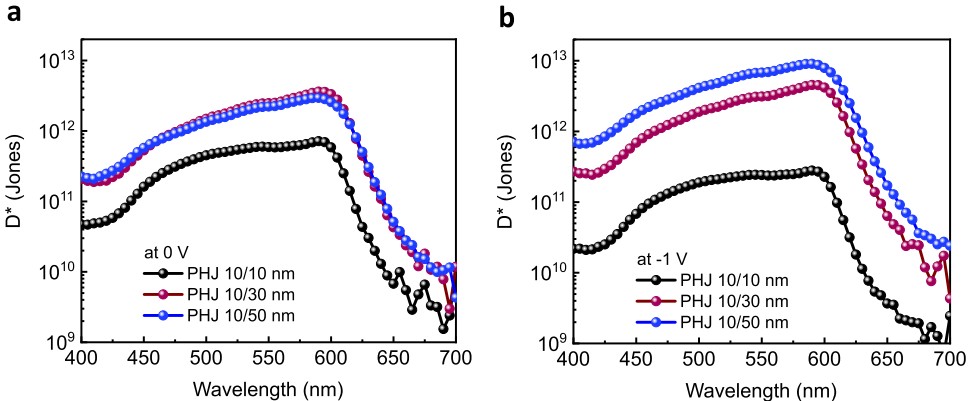

**Fig. 7 | Thickness-dependent photodetector performance under bias.** Specific detectivity ($D^*$) versus wavelength under different biases calculated from responsivity at (**a**) 0 V and (**b**) −1 V for all PHJ OPDs.

of the efficient direct charge generation we report in the $Cl_6$-SubPc OPD's studied herein.

## Photodetector performance

We conclude by evaluating the responsivity and detectivity of MPTA/$Cl_6$-SubPc OPDs under different biases. The responsivity for all PHJ devices is determined from the relation $R = EQE\frac{q\lambda}{hc}$ at short-circuit and at −1 V, as shown in Supplementary Fig. 16a, b where $q$ is electronic charge, $\lambda$ is light wavelength, $h$ is Planck's constant and $c$ is speed of

light. It is apparent that the responsivity for the thickest device (PHJ 10/50 nm) is more than twice as compared to the thinnest PHJ 10/10 nm. Furthermore, the responsivity increases at negative bias (−1 V), consistent with the EQE response as shown in Fig. 2d–f. The specific detectivity ($D^*$) is shown in Fig. 7a, b and calculated from the relation $D^* = \frac{R}{\sqrt{2qJ_d}}$, assuming shot noise is the dominant contribution, where $R$ is responsivity and $J_d$ is the dark current density[48]. It is observed that the specific detectivity increases under reverse bias with a striking circa 50-fold increase with increasing $Cl_6$-SubPc thickness, peaking at

$9 \times 10^{12}$ Jones for the 10/50 nm device at $-1$ V and 590 nm. Specific detectivity analyses considering the overall measured noise are shown in Supplementary Fig. 17a, b and exhibit a similar trend between devices, with an optimum $D^* \sim 4.13 \times 10^{10}$ Jones at $-2$ V. Whilst slightly higher detectivities have been reported for bulk-heterojunction OPDs[48], the specific detectivity at $-1$ V of our 10/50 nm device is one of the highest reported detectivity to date for a planar heterojunction OPD (previously reported PHJ OPDs have $D^* = 6.5 \times 10^{10}$ Jones[1], $D^* = 2.5 \times 10^{12}$ Jones[49] and $D^* = 1.3 \times 10^{12}$ Jones[50] for ID-MeIC/$C_{60}$, PM6/IT4F and P3HT/PCBM respectively). The higher detectivity with increased $Cl_6$-SubPc layer thickness results primarily from $J_d$ suppression, correlated with a lower trap density as discussed above. We note that the transport layers used in this study were originally optimized for BHJ rather than PHJ devices as such optimisation of these layers can be expected to lead to further reduction in dark/noise current. Most strikingly, these data demonstrate that OPD devices based on direct charge photogeneration by a $Cl_6$-SubPc photoactive layer can yield high performance in OPD devices.

## Discussion

In conclusion, we have investigated MPTA/$Cl_6$-SubPc PHJ-based OPDs with fixed MPTA donor thickness (10 nm) and varying $Cl_6$-SubPc acceptor thicknesses (10 nm, 30 nm, 50 nm). Remarkably, PHJ 10/50 nm devices, which have imbalanced donor and acceptor thicknesses, show the best device performance with the lowest dark current as well as the highest photocurrent density under the reverse bias conditions. Photoluminescence, transient absorption, and EQE data indicate that photocurrent generation in PHJs with thick $Cl_6$-SubPc layers originates primarily from free charge carriers within the bulk of $Cl_6$-SubPc layer, rather than from exciton separation at the donor/acceptor interface. Our DFT simulations indicate that this free charge generation in the $Cl_6$-SubPc layer can be attributed to the high octupole moment of $Cl_6$-SubPc molecules, which is sufficiently large to modulate $Cl_6$-SubPc's HOMO/LUMO level energetics by 200 meV, generating energetic offsets to drive charge photogeneration. Thicker PHJ devices exhibit faster photo-response times than thinner PHJ devices, suggesting that $Cl_6$-SubPc does not limit charge transport in the device. Under reverse bias, thicker PHJ devices show excellent photocurrent linearity (negligible bimolecular recombination losses) and low dark currents, consistent with observed lower trap densities. Planar-heterojunction OPDs with thick $Cl_6$-SubPc layers show EQE up to 35%, detectivities up to $10^{13}$ Jones, a response time of 5 μs and narrow (<100 nm FWHM) spectral response.

Overall, this study shows that $Cl_6$-SubPc is a promising material for single-component photoactive layers in OPD applications. Most strikingly, it demonstrates the potential of high molecular octupoles to drive direct charge generation in single-component photoactive layer OPD's. This opens up a new pathway for the design of high-order moment organic molecules to achieve efficient charge generation in organic optoelectronic devices without the need for donor/acceptor heterojunctions.

## Methods

### Fabrication of devices and thin films preparation

All organic semiconductor materials were purified via sublimation under high vacuum ($<10^{-6}$ Torr) prior to use. Thin films of MPTA, $Cl_6$-SubPc and PHJs (10/10 nm, 10/30 nm, 10/50 nm) measured for absorbance and PL spectra in Fig. 1 were fabricated via thermal evaporation under high vacuum ($<10^{-7}$ Torr) at a rate of 0.35 Å s$^{-1}$ on indium tin oxide (ITO) coated glass substrates that had been cleaned with isopropyl alcohol (IPA) and acetone in an ultrasonic bath. The OPDs were fabricated on ITO-coated glass substrates by sequentially depositing the hole-extraction layer molybdenum trioxide ($MoO_x$, 30 nm), the organic PHJ layer of MPTA/$Cl_6$-SubPc (10/10 nm, 10/30 nm, and

10/50 nm), and an Al electrode (100 nm). All organic film layers were thermally evaporated under high vacuum ($<10^{-7}$ Torr). Al electrodes were thermally evaporated before the devices were finally encapsulated with glass (98.5% transmittance). The active pixel size was 0.04 cm$^2$.

### Device electrical characterization

$J$–$V$ characteristics were measured with a Keithley 2400 source under a Newport solar simulator calibrated at 100 mWcm$^{-2}$ by silicon photodiode, using neutral-density filters for light-intensity dependent measurements. A light emitting diode (LED) light source calibrated by digital Luxmeter was used for low light $J$–$V$ analysis. EQE was measured via an EQE system made of a tungsten halogen lamp coupled with a grating spectrometer (CS260-RG-4-MT-D), using a BK Precision 9110 voltage source to control applied electric field. The noise spectral density was extracted from the dark current density recorded by a Keithley 4200, followed by a fast Fourier transform. For capacitance frequency measurement Autolab (model PGSTAT-12) with NOVA 1.1 was used.

### Transient absorption spectroscopy

A broadband femtosecond TA spectrometer Helios (Spectra Physics, Newport Corp.) was used to measure TAS for thin films. Laser pulses (800 nm, 100 fs pulse duration) were generated using a 1 kHz Ti: sapphire regenerative amplifier (Solstice, Spectra Physics). One portion of the 800 nm pulses was directed to an optical parametric amplifier (TOPAS) to generate the visible pump pulses. The rest of the 800 nm pulses were routed onto a mechanical delay stage (6 ns time window) and were directed through a sapphire crystal to generate a white light probe ranging from 400 nm to 900 nm in the visible to near-infra-red region. The pump and probe beams were focused onto the same spot on the samples. During the measurements, the thin film samples were kept in a quartz cuvette under continuous nitrogen flow.

### Steady-state optical characterization

PL spectra were measured by a Renishaw in Via Raman microscope in a backscattering configuration. The samples were kept under a constant nitrogen flow in a Linkam chamber to reduce degradation effects, with laser excitation wavelengths at 514 nm (Argon, Titanium Sapphire lasers), laser spot diameter of the order of 10 μm, 25% defocused on the sample. Diffracted light was separated by a diffraction grid (300 lines per mm for PL). A Si reference sample was used for spectrometer calibration. Optimized acquisition parameters (laser power, exposure time, measurement accumulation number) were kept consistent for the same experiments. Accuracy was improved by checking the reproducibility of spectra over multiple positions on the surface. PL was also measured on encapsulated devices applying reverse bias to the electrodes. Transmittance (T) of thin films was measured by a Shimadzu UV−2550 *UV−Vis* spectrophotometer, converting it into absorbance (A) and removing the substrate contribution by the equation $A = \log\left(\frac{T_{substrat}}{T_{sample}}\right)$, with the approximation of no reflectivity.

### Transient optoelectronic measurement

The transient optoelectronic measurement was taken on an in-house built setup consisting of an oscilloscope (Tektronix TDS 3032B) connected to the computer via NI-PCI-6251 DAQ card and a ring of 12 white LEDs (Luxeon Star/O (LXHL-NWE8)) connected to a remote-controlled power supply. These LEDs turn on time is 100 ms to allow DUT to achieve steady state condition with a switch-off time of 100 ns. Charge extraction transients were recorded over 50 Ω resistance. To reduce the noise at low light levels in current transient DLPCA-200−FEMTO current amplifier was used. Transient photocurrent was measured via a DAQ card connected to a Tektronix TDS3032B Oscilloscope, illuminating the devices by a green LED (523 nm) with tunable intensity.

## Energetic characterization

Materials HOMO energy levels were measured for thin films on grounded ITO substrates using an APS04 photoemission system (KP Technology) in ambient conditions with a 2 mm gold tip. Reproducibility check and experimental error calculation were carried out by measuring at multiple positions on the surface. APS data were analysed according to the protocol described by Baikie et al.[51], extracting the HOMO from the crossing between the linear fit of the photoemission intensity cube root and the zero-emission baseline.

## Simulation

DFT simulations were performed by Gaussian09 software on the Imperial College High Performance Computing service, using GaussView 6 graphical interface for result visualizing. DFT was applied at the B3LYP level with the 6-311G (d,p) basis set, optimizing single-molecule structures in the gas phase to the minimum energy conformation and extracting octupole moment values. Electrostatic potential maps were simulated according to the Merz–Singh–Kollman (MK) model. The distributions of HOMO/LUMO energy were obtained by DFT calculations for cluster structures extracted from the amorphous morphology which was generated by the MD simulation. The MD simulation was performed for the 1000 of $Cl_6$-SubPc molecules randomly placed in the simulation cell with a force field of Optimized Potentials for Liquid Simulations (OPLS) under the NPT ensemble. Here, the temperature of the MD simulations gradually decreased from 450 K to 300 K for 4 ns and 200 clusters including central and surrounding $Cl_6$-SubPc molecules were extracted from the morphology after 10 ns of MD steps. The surrounding molecules were selected within the cut-off distance of 4 Å on the nearest neighbour distances from the central $Cl_6$-SubPc. The HOMO/LUMO energies of central $Cl_6$-SubPc in the clusters with/without partial charges of surrounding molecules were obtained by the DFT calculation with B3LYP/6-31 G(d,p). The distribution maxima of each HOMO/LUMO distribution were determined by Gaussian curve fittings.

## Data availability

The data supporting the findings of this study are available from the corresponding authors upon request.

## Code availability

The codes or algorithms used to analyse the data reported in this study are available from the corresponding authors upon request.

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

## Acknowledgements
We gratefully acknowledge Samsung Electronics for the project funding. J.D. and J.K. acknowledge UK EPSRC for ATIP programme grant (EP/T028513/1) and Centre for Doctoral Training in Plastic Electronics (EP/L016702/1).

## Author contributions
A.R., S.Y.P., C.L., and Y.D. did an optical and optoelectronic measurement on device. E.Y., D.N., and N.G. did device characterization. F.F., S.Y., J.P., J.S., and D.M. did device fabrication and modelling. K.P., J.K. and J.D. conceived the idea and proposed the project. All contributed to writing the manuscript.

## Competing interests
The authors declare no competing interests.
