## [Peer Review File · Nature Communications]

Octupole Moment Driven Free Charge Generation in Partially Chlorinated Subphthalocyanine for Planar Heterojunction Organic PhotodetectorsREVIEWER COMMENTS

Reviewer #1 (Remarks to the Author):

Dear Authors and Editor:

The manuscript by Rana et al. on "Octupole Moment Driven Free Charge Generation in Partially Chlorinated Subphthalocyanine for Planar Heterojunction Organic Photodetectors" has presented a novel study of leveraging octupole moment to facilitate photogeneration in the bulk of organic semiconductor in planar heterojunction. This can be an alternative to overcome the disorder problems in bulk heterojunction. The study has good analysis, but there are some points below to clarify and support the arguments:

1. The authors claim lower trap density is the origin of lower dark current. The dark current in Fig 2b should be plotted with x-axis in electric field accounting for the 5x difference in field due to thickness change. If Fig 2b has accounted for the electric field, is it really still much lower dark current, or they are more similar and not really a significant point? Also in the supplement Figs S2 and S3, the error bars are overlapping between 10 and 50 nm. It doesn't seem to show significant difference in trap density.
2. For the above claim of lower trap density in thicker film, it seems to contradict that there needs to be some grain boundaries (Fig S12) to get the energy distribution to be favorable for charge separation. Please would you elaborate on this point, and why EQE is higher in thick film if energy disorder is lower (point 1 above), but then the simulations say it needs that grain boundary to facilitate charge generation. A minor point is there is a typo of the phrase "parital chargers" in the last sentence of the caption of Fig S12.
3. The response speed is surprising, with slower transport time for thinner devices. This was attributed to trap density again. But I think it would be more convincing if the authors present data of Cap-freq under light, and then the extracted trap densities will reflect this photoresponse. Perhaps the cap vs freq drop will shift to higher freq with thicker film. The fig S1 is only in the dark right? With changing Fermi level under light, that might be not applicable to photo results (ie, transit time).

Reviewer #2 (Remarks to the Author):

The manuscript reports a study of organic photodetectors. The authors measure planar heterojunction OPD devices with varying thickness of Cl6-SubPc and propose single component exciton splitting. The overall performance seems very good. They claim that free charge generation in the Cl6-SubPc layer was observed and that this is caused by the high octupole moment of the molecules, which can modulate the energy levels by 200 meV. I am not sure about the level of novelty and significance of this work for Nature Communications.

Devices with SubPc, SubNc and halogenated variants as photoactive layers have been demonstrated earlier. Electrostatic multipole moments have also been discussed frequently in the past in connection to band structure engineering of organic semiconductors and for donor-acceptor interfaces for example.

The strongest argument presented here is: "The potential importance of molecular octupole moments determining a short-range electrostatic potential in optimising OPD performance has not been reported previously." It remains unclear if this is just a singular observation or has some potential for future developments.

I also have additional concerns. To my knowledge Cl6SubPC has also a quadrupole moment and a dipole moment. The authors argue that the quadrupole moment is small but

the distance dependence is stronger and more long range, so I think this argument is not sufficient to exclude quadrupole moments but simulations should be done. Moreover, the dipoles can be large which is reflected in the Fig. 6a. Why has this been neglected? According to simulations, Cl6-SubPc exhibits a large octupole moment. Is this really so unusual? What are typical values of other compounds e.g. other NFAs? The authors calculate a very large HOMO FWHM of 440 meV. This kind of electronic disorder is due to the polar molecules. The LUMO bandwidth can be expected to be similar. The capacitance measurements, however do not show such broad distribution and large concentration of trap states that would be consistent with the simulations. In contrast authors observe a comparatively low trap density at low energy. Exciton diffusion length has not been measured and it is not clear if this is below the layer thickness. This seems to be an important point for their reasoning. TAS is not supporting the hypothesis in my view because the charges seem not directly generated from excitons. In my opinion, there is an imbalance between the data presented and the relevant conclusions that can be reliably drawn from this. However, I am not sure if this could be fixed by a better selection or better structure.

Reviewer #3 (Remarks to the Author):

The authors study an organic photodiode based on a planar heterojunction. Within the used acceptor material, Cl-SubPc, they observe direct charge generation, and they attribute this to a very high octupole moment in the molecule.

The

- Which layer stack is used for the absorption data of fig 1(c) and the PL data of fig 1 (d)? Is this with or without contact layers?
- Why is the absorption data normalized? Some very interesting data can be gathered by looking at the absolute absorption, and comparing this with the EQE. From this comparison, a sense of the internal quantum efficiency can be obtained.
- PL data: the authors claim that the PL increases for thin bilayers, but the fig 1(d) says otherwise. The PL for the neat Cl-SubPc is higher compared to the PHJ 10/10 structure, this looking at the data at 650nm (PL peak for Cl-SubPc). The PL indeed goes up a little for the other two PHJs, but this might be attributed to optical interference phenomena. The authors also claim that this data is consistent with reported exciton diffusion lengths around 17nm, but 17nm is not negligible compared to the used layer thicknesses. If this diffusion length is correct, it would mean that +- half of the excitons can be quenched by the D/A interface. Also, depending on the used layer stack, the metal contacts will quench excitons. It is unclear from the current text if metal contacts are present for this experiment. If the Al layer is present for all stacks, the quenching behaviour has a huge impact on the conclusions. More explanation is required.
- Concerning the EQE spectra, the authors did not take optical interference into account. The Al metal that is directly in contact with Cl-SubPc will act as a mirror for the light, and the reflected light will interfere with the incoming light. This interference pattern will show peaks at a distance of the wavelength divided by 4 and divided by the refractive index of the studied material. Considering the peak absorption around 600nm and assuming a refractive index of around 2, this will lead to a peak interference at +- 75nm away from the Al contact. As such it is not surprising to see an increase in the EQE when using thicker Cl-SubPc layers, assuming that the photogeneration is due to exciton dissociation at the D/A interface.

The conclusion in this part of the paper that the photocurrent is due to direct photogeneration instead of exciton dissociation at the D/A interface is postulated without real evidence.

- Bias dependent PL: the bias dependency of the CI-SubPc PL (650nm) and MPTA PL (750nm) show the same trend. The text in the article (if correctly understood from this reviewer) claims that the PL data of MPTA should go down with voltage, while the PL data of SubPc should go up. I don't see this in Figure 3.

- Octupole calculation: the outcome of the DFT calculations is that local energy offsets in the range of 100-200meV are present within the SubPc layer. The authors claim that this is large enough to drive exciton dissociation within the SubPc. What is the typical exciton binding energy in this material?

While very interesting, the article requires some mandatory changes to address the aforementioned points.

Dear Reviewers,

We thank you for your valuable feedback on our manuscript titled “**Octupole Moment Driven Free Charge Generation in Partially Chlorinated Subphthalocyanine for Planar Heterojunction Organic Photodetectors**, (NCOMMS-23-41572)”. In response to the reviewers’ comments, we have carefully considered each remark and made the necessary revisions to address the concerns raised as described in the document below.

The comments and points of the referees are copied in red color and our response is colored in black. Text modifications in revised manuscript as per the referee’s comments are highlighted in yellow, accompanied by the respective page numbers and line numbers.

We hope that we satisfactorily addressed all points and appreciate for your consideration.

Yours Sincerely,

James R. Durrant and Co-Authors

Responses to the reviewer comments

Reviewer #1 (Remarks to the Author):

The manuscript by Rana et al. on “Octupole Moment Driven Free Charge Generation in Partially Chlorinated Subphthalocyanine for Planar Heterojunction Organic Photodetectors” has presented a novel study of leveraging octupole moment to facilitate photogeneration in the bulk of organic semiconductor in planar heterojunction. This can be an alternative to overcome the disorder problems in bulk heterojunction. The study has good analysis, but there are some points below to clarify and support the arguments:

1.1. The authors claim lower trap density is the origin of lower dark current. The dark current in **Fig 2b** should be plotted with x-axis in electric field accounting for the $5\times$ difference in field due to thickness change. If **Fig. 2b** has accounted for the electric field, is it really still much lower dark current, or they are more similar and not really a significant point? Also, in the supplement **Figs. S2** and **S3**, the error bars are overlapping between 10 and 50 nm. It doesn't seem to show significant difference in trap density.

Our response: Thank you for your valuable feedback on our manuscript. We appreciate the insightful comments and have thoroughly considered the points raised. Regarding the plot of dark current in **Fig. 2b**, we acknowledge the usefulness of accounting for the difference in electric field due to the thickness variations.

Fig. R1 Current density characteristics of PHJ MPTA/Cl₆-SubPc photodiodes with different thicknesses of Cl₆-SubPc under dark conditions with respect to the electric field. (Now included as **Supplementary Fig. 4**)

To address this concern, we have replotted dark currents with the x-axis presented in terms of electric field, considering the 5 times difference in field resulting from changes in thickness as depicted here in **Fig. R1** (and now as **Supplementary Fig. 4**). This new plot also shows that the thin device (PHJ 10/10nm) has over an order of magnitude higher dark current even at low reverse electric field (-0.5 MV/cm) as compared to the thick devices (PHJ 10/50 and 10/30 nm), consistent with the higher trap density of the 10/10 nm device. We note that the significantly higher dark current in the thin device at large reverse electric field is likely to be dominated by diode breakdown due to direct tunnelling of charge carriers¹. We also acknowledge that the significant error bars overlap in our photoemission spectra as shown in **Supplementary Fig. 2**, although the trend looks clear. This trend is moreover consistent with higher trap density in overall energy range from 0.50-0.65 eV for thin devices, analysed from capacitance frequency as shown in **Supplementary Fig. 3**. We further highlight that the thinnest devices also exhibit the slowest photocurrent decays as shown in **Supplementary Fig. 10** in (specifically a large slow decay phase), again consistent with a higher trap density. Together we believe these data are all indicative of lower trap densities in the thicker devices contributing to the smaller dark current as compared to thin devices.

In response to the referee's comments, we have inserted the following text on page 9 line 184 to 188 in revised manuscript.

'We also plotted the dark current versus electric field (accounting for the difference in field with organic film thickness), as shown in **Supplementary Fig. 4**; in this plot the dark current is still highest (by over an order of magnitude difference at low field) for the thinnest device. All of these analyses are indicative of a lower trap density in thickest device, which is likely to contribute to the lower dark current in this device.'

1.2. For the above claim of lower trap density in thicker film, it seems to contradict that there needs to be some grain boundaries (**Fig. S12**) to get the energy distribution to be favourable for charge separation. Please would you elaborate on this point, and why EQE is higher in thick film if energy disorder is lower (point 1 above), but then the simulations say it needs that grain boundary to facilitate charge generation. A minor point is there is a typo of the phrase "parital chargers" in the last sentence of the caption of **Fig. S12**.

Our response: We thank to the reviewer for insightful comment on the potential relationship between trap density and grain boundaries. In our previous study, we have concluded that the electrostatic landscape generated by the differently packed α -6T molecules (e.g., face-on vs edge-on) can create 400 meV energetic offset which is large enough for the efficient exciton separation². For Cl₆-SubPc, the molecular origin of energetic offsets required for efficient charge generation was not clear. To check if the energetic offset originated from differences in local molecular packing or the grain boundaries, we performed the molecular dynamics simulations. First, the randomly packed amorphous structure was generated for 1000 molecules of Cl₆-SubPc. In this structure, we observe a large 200 meV HOMO level shift when the partial charges were included, assigned to the high octupole moment of Cl₆-SubPc molecules surroundings. In contrast, the energetic offset induced by the partial charges at the grain boundaries was only 45 meV, which is unlikely to be large enough for efficient exciton separation. As already stated in the main text that this simulation result confirms that (line 355-357) *for Cl₆-SubPc, the largest partial charge induced energetic shifts result mainly from differences in local molecular packing rather between bulk and grain boundaries, consistent with expected shorter range of octupole interactions.* These results also confirms that any charge photogeneration in Cl₆-SubPc is likely to be an intrinsic property of the Cl₆-SubPc (i.e., its high octupole moment) rather than a result of applied electric fields consistent with no PL quenching under applied electric fields.

In summary, the higher EQE observed in thicker film may result from more *differences in local molecular packing*, not due to higher grain boundaries. The lower trap density in thicker film will assist efficient charge transport and extraction. Therefore, there is no contradiction implying that there needs to be some grain boundaries in thicker film for its high EQE. In contrast, thick films can have higher EQE due to less traps leading to reduced recombination and better charge collection.

In addition, we have rectified typographical error in the caption of **Supplementary Fig. 12** (now as **Supplementary Fig. 14**). In response to the reviewer comment above, the following text has also added in main manuscript on page 15 line 357 -362 as follows.

“These results also confirms that any charge photogeneration in Cl₆-SubPc is likely to be an intrinsic property of the Cl₆-SubPc (i.e., due to its high octupole moment) rather than a result of applied electric fields, consistent with no PL quenching under applied electric fields. We note that the octupole induced energetic shifts may also impact upon charge transport, dependent upon film nanomorphology, a topic beyond the scope of this study.”

1.3. The response speed is surprising, with slower transport time for thinner devices. This was attributed to trap density again. But I think it would be more convincing if the authors present data of Cap-freq under light, and then the extracted trap densities will reflect this photoresponse. Perhaps the cap vs freq drop will shift to higher freq with thicker film. The **Fig. S1** is only in the dark right? With changing Fermi level under light, that might be not applicable to photo results (i.e., transit time).

Our response: We appreciate your suggestion to include data on capacitance frequency under light conditions to further strengthen the correlation between transport time and trap densities, which could offer more convincing evidence. The presentation of capacitance frequency response under illumination has indeed provided a clearer insight into the transient photo response and its direct reflection on the extracted trap densities. For comparison, normalized capacitance frequency response under light is illustrated in **Fig. R2a**, distinctly showing the frequency drop is shifting towards higher frequencies with increased film thickness. Upon taking into account a dielectric constant of 4.92 (**Supplementary Fig. 8**) and 50Ω resistance, the calculated RC time constants for these devices are $0.40\ \mu\text{s}$, $0.2\ \mu\text{s}$ and $0.14\ \mu\text{s}$ respectively. This explicitly indicates that the transient current and trap response is not limited by the RC factor.

Fig. R2b shows trap density with respect to the energy extracted from the capacitance frequency response under light condition. The trap density under light (**Fig. R2b**) shows a similar trend with higher trap density in the thinnest device as observed in the dark (**Fig. 2c** in the main manuscript).

Fig. R2 a Normalized capacitance response with respect to frequency at $10\ \text{mW}/\text{cm}^2$ white light intensity **b** extracted trap density with respect to energy measured from capacitance frequency measurement under light.

Therefore, the slower response speed for thinner devices may indicate that there can be more significant contributions from interfaces (e.g., between the photoactive layer and charge extraction layers) and local molecular packing leading to trap formation. To support our transient response analysis, as suggested by the referee, we have further added capacitance frequency response under light in the supporting information as **Supplementary Fig. 11** including text details in main manuscript on page 13 line 285 -287 as follows.

“As the thickness increases, the illuminated capacitance response drop shifts to higher frequencies, indicating that the thick devices have a faster response, consistent with their transient photoresponse, as shown in **Supplementary Fig. 11**”

Reviewer #2 (Remarks to the Author):

2.1. The manuscript reports a study of organic photodetectors. The authors measure planar heterojunction OPD devices with varying thickness of Cl₆-SubPc and propose single component exciton splitting. The overall performance seems very good. They claim that free charge generation in the Cl₆-SubPc layer was observed and that this is caused by the high octupole moment of the molecules, which can modulate the energy levels by 200 meV. I am not sure about the level of novelty and significance of this work for *Nature Communications*.

Our response: Thank you for your comment. We appreciate your interest in our work and your constructive feedback. We would like to address your concern about the novelty and significance of our work for *Nature Communications*.

We believe that our work presents a novel and significant contribution to the field of organic photodetectors (OPDs) for the following reasons:

The key novelty of this paper is that we demonstrate for the first time that a high molecular octupole moment can result in direct charge generation within an organic single component film. This is a remarkable property of Cl₆-SubPc (octupole moment $-80 \text{ D}\text{\AA}^2$) that distinguishes it from other known organic semiconductors and enables efficient exciton splitting without the need for a heterojunction. We support this claim by optical and optoelectronic analyses, as well as molecular modelling. This novelty is also highlighted by reviewer 1 who stated that “*this work has presented a novel study of leveraging octupole moment to facilitate photogeneration in the bulk of organic semiconductor in planar heterojunction*”. We anticipate that the current

study will pave the way for the exploration of innovative approaches in designing efficient optoelectronic materials and devices, leveraging higher-order molecular moments.

We demonstrate that the thickness of the Cl₆-SubPc layer can be tuned to optimize the OPD performance, and achieve detectivities approaching $\sim 10^{13}$ Jones, with a dark current below 10^{-7} A cm⁻² up to -5 V. These values are comparable to or even better than those reported for the state-of-the-art OPDs based on BHJ or PHJ structures.

We hope that these points clarify the novelty and significance of our work and justify its publication in Nature Communications. We thank you again for your valuable feedback.

To highlight the novelty of this study, we have revised the final sentences of our abstract and conclusion as follows.

Abstract “Based on these findings, we conclude that high octupole moment molecular semiconductors are promising materials for high-performance OPDs employing a single-component photoactive layer”.

Conclusion “Most strikingly, it demonstrates the potential of high molecular octupoles to drive direct charge generation in single component photoactive layer OPD’s. This opens up a new pathway for the design of high order moment organic molecules to achieve efficient charge generation in organic optoelectronic devices without the need for donor / acceptor heterojunctions.”

2.2. Devices with SubPc, SubNc and halogenated variants as photoactive layers have been demonstrated earlier. Electrostatic multipole moments have also been discussed frequently in the past in connection to band structure engineering of organic semiconductors and for donor-acceptor interfaces for example. The strongest argument presented here is: “The potential importance of molecular octupole moments determining a short-range electrostatic potential in optimising OPD performance has not been reported previously. “It remains unclear if this is just a singular observation or has some potential for future developments.

Our response: Following the answer to the previous comment, we note that large octupole moments are unlikely to be limited to Cl₆-SubPc. Indeed, we have already identified another SubPc with an even higher octupole moment and even more efficient charge generation, although this will be the subject of a follow up study. As such this study raises the possibility

of a new approach to design molecular materials for organic optoelectronic devices. Therefore, we think that our study is not just a singular observation but has great potential for future developments. We hope that our work can inspire more research on the role of molecular octupole moments in OPD and other organic optoelectronic devices. We also hope that our work can provide a useful guideline for the rational design of new OPD materials with high performance and low cost.

2.3. I also have additional concerns. To my knowledge Cl₆-SubPC has also a quadrupole moment and a dipole moment. The authors argue that the quadrupole moment is small, but the distance dependence is stronger and more long range, so I think this argument is not sufficient to exclude quadrupole moments but simulations should be done. Moreover, the dipoles can be large which is reflected in the Fig. 6a. Why has this been neglected?

Our response: We thank the reviewer for bringing up this important point. We carefully considered the role of both dipole and quadrupole moments in influencing energetics and intermolecular interactions in the studied systems. In terms of dipole moments, our DFT calculations show that the cone-shaped Cl₆-SubPc molecules have a similar dipolar character to other NFAs commonly used in organic photodiodes for which dipole moments are typically very low and considered having a negligible effect on the optoelectronic characteristics (e.g., 1.52 Debye for Cl₆-SubPc vs 1.02 Debye for Y6). For comparison, molecules designed specifically with a dipolar character have much higher dipoles, such as 4.06 Debye for MPTA. As for quadrupoles, we included in the Supplementary Information (**Supplementary Fig. 12**) the simulated values of Q_{π} for SubPc-type molecules ($\sim 15 \text{ ea}_0^2$), which are substantially smaller as compared to common acceptors where quadrupolar character plays a significant role ($> 200 \text{ ea}_0^2$ for Y6 and IT-4F, as shown in our recent paper in Nature communications³).

In response to reviewer comment we have added following text in main manuscript on page 14 line 307 -310 with our recent nature comm. paper as reference 44:

“However, as shown in **Supplementary Fig. 12**, Cl₆-SubPc exhibits a negligible (>10 times smaller) quadrupole moment, compared to common acceptor molecules, ruling this out as a potential origin of the observed efficient direct charge generation”.

44. Fu, Y. et al. Molecular orientation-dependent energetic shifts in solution-processed non-fullerene acceptors and their impact on organic photovoltaic performance. *Nature Communications* **14**, 1870 (2023).

2.4. According to simulations, Cl₆-SubPc exhibits a large octupole moment. Is this really so unusual? What are typical values of other compounds e.g., other NFAs?

Our response: Following from what discussed in the response to the previous question, our DFT simulations show that octupole moments in Cl₆-SubPc are orders of magnitude higher than typical NFAs as shown in **Fig. R3** (**Fig. 6a**) in the revised main manuscript) e.g., the octupole moment is close to 0 for Y6, which led us to conclude that the dominant role in influencing molecular energetics is held by this term in the higher order multipole expansion.

Fig. R3. Calculated Octupole moment in Cl₆-SubPc as compared to the donor molecule (MPTA) and other common NFAs as taken from **Fig. 6a** in revised main manuscript.

2.5. The authors calculate a very large HOMO FWHM of 440 meV. This kind of electronic disorder is due to the polar molecules. The LUMO bandwidth can be expected to be similar. The capacitance measurements, however, do not show such broad distribution and large concentration of trap states that would be consistent with the simulations. In contrast authors observe a comparatively low trap density at low energy.

Our response: Thank you for your insightful observation regarding our work. Indeed, our calculation show gaussian function with 440 meV of HOMO distribution with amorphous structure (**Fig. 6b**) in the revised manuscript) which is likely to occur due to the high different local molecular packing in the film. For Cl₆-SubPc, a similar gaussian distribution of 400meV determined by photoelectron spectroscopy has also reported by H. Lee et. al.⁴ We agree that HOMO/LUMO distribution partly depends on morphology, however, the main purpose of our simulation was to see the overall effect of partial charges on the molecular energetics. The close alignment of HOMO level maxima from simulation with partial charges with that determined from photoemission data (**Fig. 1f** in the revised manuscript) further validates our calculation.

Additionally, we would like to point out that capacitance frequency measurement reflects the density of deep trap states, where the frequency determines the fraction of trapped charges responsive to the AC field through thermal excitation. The small AC applied voltage shifts the Fermi energy, releasing carriers from states near the Fermi level. At a specific frequency, only states shallower than the demarcation energy (E_t) can respond quickly enough to contribute to capacitance. The derivative $(\frac{\omega}{kT}) \frac{dC}{d\omega}$ in eq.2 (see supplementary information) isolates the response of traps within kT of E_t . Therefore, capacitance in the dark does not correspond to the HOMO/LUMO energy distribution but reflects the distribution of localized tail states extending into the bandgap. This may explain the significant difference in Full Width at Half Maximum (FWHM) between the Gaussian distribution in our DFT simulation data and the trap density data, although a full analysis of this is beyond the scope of our study.

2.6. Exciton diffusion length has not been measured and it is not clear if this is below the layer thickness. This seems to be an important point for their reasoning.

Our response: We express our thanks to the reviewer for raising a reasonable concern. Typically, the exciton diffusion length for organic materials falls within the range of 3-20 nm⁵, which is considerably shorter than the optical absorption length. This limitation restricts exciton migration to the interface. The exciton diffusion length of SubPc molecules has been extensively investigated by various research groups. T. Zhang et al.⁶ reported a SubPc exciton diffusion length of 16.6 nm, while H. Gommans et al.⁷ found a diffusion length of 28 nm through photoluminescence (PL) quenching experiments. Richard R. Lunt⁸, using a similar method, reported an 8 nm diffusion length. W. A. Luhman et. al.⁹ and S. M. Menke et. al.¹⁰

also reported SubPc diffusion length consecutively 7.7 nm and 10.7 nm. Different values of diffusion length likely to be caused by diversity in approaches/methods used for estimation. Collectively, these findings consistently indicate that the exciton diffusion length is shorter than the SubPc layer thickness in thick OPD devices (PHJ 10/50 nm). As a result, we infer that photocurrent generation in these devices is primarily due to the electrostatic potential induced energetic offset imposed by high molecular octupole moment of Cl₆-SubPc, not the result of exciton separation at the MPTA/Cl₆-SubPc interface.

In order to clarify this point, the text on page 5 line 120 -123 have been modified to say:

“This is consistent with reported exciton diffusion lengths of SubPc’s of 7-28 nm, shorter than the layer thickness such that photogenerated excitons are unable to reach the D/A interface. Consistent with this conclusion, PL from Cl₆-SubPc is quenched in case of PHJ 10/10 nm, as 10 nm of Cl₆-SubPc layer is shorter than its reported exciton diffusion length.”

2.7. TAS is not supporting the hypothesis in my view because the charges seem not directly generated from excitons.

Our response: We think the referee may have misunderstood the conclusion we draw from our TAS data. The key conclusion from our TAS data is that at long times, after exciton decay, we observe clear spectral signatures of charges even for a neat Cl₆-SubPc film, indicative of charge generation within this film. This conclusion is independent of whether these charges are formed directly following photoexcitation or result from exciton decay on the picosecond (ps) timescale. The referee is correct that our kinetic deconvolution indicate that these charges are formed within our instrument response, suggesting a branching between exciton and charge generation directly following photoexcitation. In our study, we do not address the details of exciton diffusion, nor how this might be impacted by charge generation from these excitons. It is possible that direct charge generation may result from photoexcitations directly adjacent to large octupole generated molecular energetic offsets. A comment on this has been added on page 17 line 362 – 364 in the revised manuscript.

“It is likely that the direct (instrument response limited) charge generation indicated by our transient absorption data results from photoexcitations occurring in regions where the local molecule packing results in large octupole induced energetic offsets.”

2.8. In my opinion, there is an imbalance between the data presented and the relevant conclusions that can be reliably drawn from this. However, I am not sure if this could be fixed by a better selection or better structure.

Our response: We agree that ensuring a more coherent connection between the data and the conclusions is crucial for the overall strength of the paper. In our revision, we have carefully reassessed the presentation and organization of the data to enhance its alignment with the conclusions drawn. This have been achieved more strategic selection and additional inclusion of data sets, as well as a clearer articulation of the logical progression between the information presented and the resulting conclusions. For example, we have added capacitance frequency response in light, field dependent dark current analysis to further support our conclusion drawn from trap dependent transient photo current response. We have also made required text modifications as suggested by reviewer to enhance the clarity of our conclusions.

Reviewer #3 (Remarks to the Author):

The authors study an organic photodiode based on a planar heterojunction. Within the used acceptor material, Cl₆-SubPc, they observe direct charge generation, and they attribute this to a very high octupole moment in the molecule.

3.1. Which layer stack is used for the absorption data of fig 1(c) and the PL data of fig 1 (d)? Is this with or without contact layers?

Our response: The thin films used in **Fig. 1c, d** were prepared on top of ITO coated glass substrates. We have amended the Methods section on page 19 line 425 -427 in the revised manuscript as shown below.

Thin films of MPTA, Cl₆-SubPc and PHJs (10/10, 10/30, 10/50 nm) measured for absorbance and PL spectra in **Fig. 1** were fabricated via thermal evaporation under high vacuum (<10⁻⁷ Torr) at a rate of 0.35 Å/s on ITO coated glass substrates that had been cleaned with isopropyl alcohol (IPA) and acetone in an ultrasonic bath.

3.2. Why is the absorption data normalized? Some very interesting data can be gathered by looking at the absolute absorption and comparing this with the EQE. From this comparison, a sense of the internal quantum efficiency can be obtained.

Our response: We thank the reviewer to raise this important point. We compared non-normalized absorption spectra and EQE spectra measured with no bias vs. bias at -5 V, as shown in **Fig. R4**. Absorbance increases linearly as a function of total thickness of PHJs (**Fig. R4d**, whereas EQE doesn't show linear dependence on the total thickness (**Fig. R4 e,f**). If we assume that any optical interference effect isn't considered at this stage (reasonable assumption for the thicknesses studied in this work), namely absorbance here indicates the number of photons incident to photoactive layer, the ratio of EQE difference (~53%) between 10/10 and 10/30 nm PHJ is much larger than the ratio of absorbance difference (26%), regardless of bias condition. This can imply that thicker PHJ devices (10/30 and 10/50 nm) have higher IQE compared to the thinnest condition (10/10 nm), consistent with the lower trap densities in thicker films, although full analysis of this would require detailed consideration of optical interference effects in these devices.

Fig. R4. **a** Absorbance spectra of PHJ films on ITO/glass substrates. EQE spectra of PHJ devices with **b** no bias and **c** voltage at -5 V **d** Cl₆-SubPc thickness dependent absorption at 590 nm **e** thickness dependent EQE at 590 nm for 0V and -5 V **f** thickness dependent normalized EQE for 0V and -5 V.

3.3. PL data: the authors claim that the PL increases for thin bilayers, but the Fig. 1d says otherwise. The PL for the neat Cl₆-SubPc is higher compared to the PHJ 10/10 structure, this

looking at the data at 650nm (PL peak for Cl-SubPc). The PL indeed goes up a little for the other two PHJs, but this might be attributed to optical interference phenomena. The authors also claim that this data is consistent with reported exciton diffusion lengths around 17nm, but 17nm is not negligible compared to the used layer thicknesses. If this diffusion length is correct, it would mean that +/- half of the excitons can be quenched by the D/A interface. Also, depending on the used layer stack, the metal contacts will quench excitons. It is unclear from the current text if metal contacts are present for this experiment. If the Al layer is present for all stacks, the quenching behaviour has a huge impact on the conclusions. More explanation is required.

Our response: We thank the reviewer to point out an unclear explanation for PL data in the original version of the manuscript. The PL data is obtained from PHJ films deposited on glass/ITO substrates (see **Fig. 1d**) without Al layer as mentioned in methods section. The quenching of the photoluminescence (PL) peak at 650 nm in a 10/10 nm film of Cl₆-SubPc can be explained by the exciton diffusion length, which is approximately 17 nm, as 10 nm of Cl₆-SubPc layer thickness is shorter than its exciton diffusion length. The other PHJ samples (10/30 and 10/50 nm) did not show any strong PL quenching (even increased especially for 10/30 nm case), unlike typical BHJs. This could be attributed to very limited D/A interfacial area in PHJ samples, indicating PL quenching at D/A interface is not the major origin of photocurrent generation in the devices. The referee is correct that the presence of the addition of the metallic Al contact (as in **Fig. 3**) may cause interference effects as well as PL quenching at this contact. This is likely to explain the different trends in PL intensity between the three devices with Al in **Fig. 3** compared to the films in **Fig. 1d**. A sentence stating this has been added, as detailed below. For this reason, our analysis of the PL quenching between films was conducted in the absence of Al, using the data shown in **Fig. 1d**.

In response to the reviewer's comment, we have added additional description in the revised manuscript on page 5, line 121 – 123.

“Consistent with this conclusion, PL from Cl₆-SubPc is quenched in case of PHJ 10/10 nm, as 10 nm of Cl₆-SubPc layer is shorter than its reported exciton diffusion length.”

and on page 9, lines 201-204 as follows.

“We note the PL intensities of these devices are likely to be impacted by interference (and possible quenching) effects resulting from the presence of the Al top contact, as such we focus here only on the bias dependence of this device PL.”

3.4. Concerning the EQE spectra, the authors did not take optical interference into account. The Al metal that is directly in contact with Cl₆-SubPc will act as a mirror for the light, and the reflected light will interfere with the incoming light. This interference pattern will show peaks at a distance of the wavelength divided by 4 and divided by the refractive index of the studied material. Considering the peak absorption around 600nm and assuming a refractive index of around 2, this will lead to a peak interference at +/- 75nm away from the Al contact. As such it is not surprising to see an increase in the EQE when using thicker Cl₆-SubPc layers, assuming that the photogeneration is due to exciton dissociation at the D/A interface. The conclusion in this part of the paper that the photocurrent is due to direct photogeneration instead of exciton dissociation at the D/A interface is postulated without real evidence.

Our response: The referee is correct that optical interference effects may also enhance light absorption for the 10/50 nm device and may indeed be a factor in the increased EQE for this device. However, this does not change the argument that a 50 nm Cl₆-SubPc layer is thicker than its exciton diffusion length, such that in the absence of charge generation within this layer, this enhanced light absorption would not be expected to result in a higher EQE. Indeed, there are numerous reports of PHJ organic devices, where increasing the light absorbing layer thickness >10 nm does not result in increased EQE (and often lower EQEs)¹¹. As such, the higher EQE we observe here for the 50 nm thick layer is remarkable, and consistent with our conclusion that charge photogeneration in this device is not dominated by charge generation at the hetero-junction but must rather result from charge generation within the 50 nm Cl₆-SubPc layer. In response to the reviewer comments, we have added the following sentence on Page 8 line 157-158.

“The thicker layer will result in higher light absorption, most likely aided by optical interference effects”.

3.5. Bias dependent PL: the bias dependency of the Cl-SubPc PL (650nm) and MPTA PL (750nm) show the same trend. The text in the article (if correctly understood from this reviewer) claims that the PL data of MPTA should go down with voltage, while the PL data of SubPc should go up. I don't see this in **Fig. 3**.

Our response: As we described in the original manuscript, in case of PHJ 10/10 nm, PL is dominated by MPTA emission at ~ 725 nm (most likely CT state emission rather than excimers based on our previous observation (in *Nat. Commun.* **13**, 3745, (2022))¹². This MPTA PL decreases by applying reverse bias (**Fig. 3a**), indicative of efficient CT state quenching. On the other hand, in case of PHJ 10/30 and 10/50 nm, PL is dominated by Cl₆-SubPc excitonic emission at 650 nm. In this case, PL is not quenched by external electric field (even slightly increased) as shown in **Fig. 3b** and **3c**, implying excitons in Cl₆-SubPc are not significantly impacted by reverse bias.

3.6. Octupole calculation: the outcome of the DFT calculations is that local energy offsets in the range of 100-200meV are present within the SubPc layer. The authors claim that this is large enough to drive exciton dissociation within the SubPc. What is the typical exciton binding energy in this material? While very interesting, the article requires some mandatory changes to address the aforementioned points.

Our response: The typical exciton binding energy in organic molecule is few hundreds of meV¹³. However, in literature Guo Li¹⁴ has reported binding 490 meV for SubPc monomer from DFT study. Whereas X. Chen¹⁵ has reported 530 meV binding energy for SubPc through modelling. However, these calculated exciton binding energy values can substantially overestimate as compared to the solid-state or liquid phase due to electronic polarization mainly originated from the induction effect of charges.¹⁶ The molecular arrangements can also play important role in regulating exciton binding energy of photoactive material. In practice, there are now many examples of organic BHJ solar cells where energetic offsets of < 300 meV are sufficient to drive almost unity efficiency exciton separation^{17,18}. As such, it appears reasonable that energetic offsets of 100-200 meV would be sufficient to drive the moderately efficient charge generation reported herein.

In our case the simulation was performed with force field of Optimized Potentials for Liquid Simulations (OPLS) under the NPT ensemble for the 1000 of Cl₆-SubPc molecules with amorphous morphology. The surrounding molecules were selected within the cut-off distance of 4Å on nearest neighbour distances from the central Cl₆-SubPc. In gas phase simulations, the molecules are not influenced by intermolecular forces, while in liquid simulations, the molecules are influenced by intermolecular forces. Thus, we have carefully reduced the

electronic polarization effect of gas phase in our energetic calculation. Therefore, we believe that 100-200 meV energy should be enough for assisting separation of excitons in Cl₆-SubPc. We have thoroughly considered your suggestion and made necessary changes to improve quality of article.

References

1. Jang, W. *et al.* Theoretical and Experimental Investigation of Barrier-Energy-Dependent Charge Injection Mechanisms in Organic Photodetectors. *Adv Funct Mater* **33**, 2209615 (2023).
2. Dong, Y. *et al.* Orientation dependent molecular electrostatics drives efficient charge generation in homojunction organic solar cells. *Nat Commun* **11**, 4617 (2020).
3. Fu, Y. *et al.* Molecular orientation-dependent energetic shifts in solution-processed non-fullerene acceptors and their impact on organic photovoltaic performance. *Nat Commun* **14**, 1870 (2023).
4. Lee, H. *et al.* Interfacial electronic structure of Cl 6 SubPc non-fullerene acceptors in organic photovoltaics using soft X-ray spectroscopies. *Physical Chemistry Chemical Physics* **19**, 31628–31633 (2017).
5. Mikhnenko, O. V., Blom, P. W. M. & Nguyen, T. Q. Exciton diffusion in organic semiconductors. *Energy Environ Sci* **8**, 1867–1888 (2015).
6. Zhang, T., Dement, D. B., Ferry, V. E. & Holmes, R. J. Intrinsic measurements of exciton transport in photovoltaic cells. *Nat Commun* **10**, 1156 (2019).
7. Gommans, H., Schols, S., Kadashchuk, A., Heremans, P. & Meskers, S. C. J. Exciton Diffusion Length and Lifetime in Subphthalocyanine Films. *Journal of Physical Chemistry C* **113**, 2974–2979 (2009).
8. Lunt, R. R., Giebink, N. C., Belak, A. A., Benziger, J. B. & Forrest, S. R. Exciton diffusion lengths of organic semiconductor thin films measured by spectrally resolved photoluminescence quenching. *J Appl Phys* **105**, (2009).
9. Luhman, W. A. & Holmes, R. J. Investigation of Energy Transfer in Organic Photovoltaic Cells and Impact on Exciton Diffusion Length Measurements. *Adv Funct Mater* **21**, 764–771 (2011).
10. Menke, S. M., Luhman, W. A. & Holmes, R. J. Tailored exciton diffusion in organic photovoltaic cells for enhanced power conversion efficiency. *Nature Mater* **12**, 152–157 (2013).
11. Ho Lee, T. *et al.* Organic Planar Heterojunction Solar Cells and Photodetectors Tailored to the Exciton Diffusion Length Scale of a Non-Fullerene Acceptor. *Adv Funct Mater* **32**, 2208001 (2022).
12. Labanti, C. *et al.* Light-intensity-dependent photoresponse time of organic photodetectors and its molecular origin. *Nat Commun* **13**, 3745 (2022).

13. Zhu, Y. *et al.* Exciton Binding Energy of Non-Fullerene Electron Acceptors. *Advanced Energy and Sustainability Research* **3**, 2100184 (2022).
14. Li, G. & Zheng, S. Exploring the effects of axial halogen substitutions of boron subphthalocyanines on the performance of BsubPC/C60 organic solar cells: a DFT/TDDFT-based computational study. *New Journal of Chemistry* **43**, 12719–12726 (2019).
15. Chen, X. & Zheng, S. On the study of influence of molecular arrangements and dipole moment on exciton binding energy in solid state. *Int J Quantum Chem* **121**, e26511 (2021).
16. Zhu, L., Tu, Z., Yi, Y. & Wei, Z. Achieving Small Exciton Binding Energies in Small Molecule Acceptors for Organic Solar Cells: Effect of Molecular Packing. *Journal of Physical Chemistry Letters* **10**, 4888–4894 (2019).
17. Sun, B. *et al.* Toward More Efficient Organic Solar Cells: A Detailed Study of Loss Pathway and Its Impact on Overall Device Performance in Low-Offset Organic Solar Cells. *Adv Energy Mater* **13**, 2300980 (2023).
18. Benatto, L., Bassi, M. D. J., De Menezes, L. C. W., Roman, L. S. & Koehler, M. Kinetic model for photoluminescence quenching by selective excitation of D/A blends: implications for charge separation in fullerene and non-fullerene organic solar cells. *J Mater Chem C Mater* **8**, 8755–8769 (2020).

REVIEWER COMMENTS

Reviewer #1 (Remarks to the Author):

I am satisfied with the revisions and response to reviewers' comments.

Reviewer #2 (Remarks to the Author):

The manuscript reports a study of organic photodetectors that are built with chlorinated subphthalocyanine as the single-component photoactive layer in devices of varying thickness.

The author highlight that despite the absence of donor-acceptor heterojunctions, the efficiency of their devices is comparable to state of the art OPD's

My strongest initial concern was that a singular material would limit the applicability of the approach of using molecules with large octupole moments. It has been confirmed by the authors that this is not the case. They have also addressed my additional remarks and requests appropriately.

The manuscript can be accepted.

Reviewer #3 (Remarks to the Author):

The authors replied to the reviewers comments. My biggest concern however, was not rightfully addressed, only in very vague terms. It is proven that optical interference has a huge impact on light absorption of these thin stacks, and this will impact both PL and EQE measurements. This will even impact PL data of PHJ stacks without a top metal, as SubPc materials tend to have a large optical index of refraction around the peak absorption. For EQE, this becomes an even higher issue, as you move the interference peak closer to / away from the D-A interface, which will influence the amount of excitons that can diffuse to this interface. Without a decorrelation of the two effects (optical interference and free-carrier absorption), I cannot assess the claimed conclusions and cannot support publication of this article in Nature Communication.

Reviewer #1 (Remarks to the Author):

I am satisfied with the revisions and response to reviewers' comments.

Our Response: Thank you for your positive feedback and acknowledgment of the revisions and responses to comments. We appreciate your thorough evaluation and are pleased that the changes have met your satisfaction.

Reviewer #2 (Remarks to the Author):

The manuscript reports a study of organic photodetectors that are built with chlorinated sub-phthalocyanine as the single-component photoactive layer in devices of varying thickness. The author highlight that despite the absence of donor-acceptor heterojunctions, the efficiency of their devices is comparable to state of the art OPD's. My strongest initial concern was that a singular material would limit the applicability of the approach of using molecules with large octupole moments. It has been confirmed by the authors that this is not the case. They have also addressed my additional remarks and requests appropriately.

The manuscript can be accepted.

Our Response: Thank you for your favourable assessment of the manuscript. We appreciate your efforts in reviewing our work. We are pleased to receive your acceptance of the manuscript.

Reviewer #3 (Remarks to the Author):

3.1 The authors replied to the reviewers comments. My biggest concern however, was not rightfully addressed, only in very vague terms. It is proven that optical interference has a huge impact on light absorption of these thin stacks, and this will impact both PL and EQE measurements. This will even impact PL data of PHJ stacks without a top metal, as SubPc materials tend to have a large optical index of refraction around the peak absorption. For EQE, this becomes an even higher issue, as you move the interference peak closer to / away from the D-A interface, which will influence the amount of excitons that can diffuse to this interface. Without a decorrelation of the two effects (optical interference and free-carrier absorption), I cannot assess the claimed conclusions and cannot support publication of this article in Nature Communication.

Our Response: We were disappointed that the reviewer did not think we had fully responded to their concerns regarding the potential impact of interference effects on our photoluminescence (PL) and external quantum efficiency (EQE) measurements. We of course

agree with the reviewer's observation that optical interference due to reflected light both in films and, more significantly, in devices may modify their PL and EQE responses, complicating analyses of data for films/devices with different thicknesses or compositions. For this reason, we had limited the conclusions we drew from these data to those which we were confident, would not be impacted by such interference effects (e.g.: we do not undertake any quantitative analysis of such data for devices as a function of film thickness). However, in response to the referee's concern, we have now collected further data on the reflectance of our devices and use these data to confirm that the conclusions we draw from our EQE and PL are not affected by the interference of reflected light.

Fig. R1 *a* Relative reflectance spectra normalised against Barium Sulfate as reflective control and *b* EQE at 0V for all PHJ OPD device stacks.

Fig. R1a plots reflectance data for the OPD devices employed in our study as a function of Cl₆-SubPc layer thickness. It is apparent that all devices show relatively low reflectivity, 0.18-0.04 with the thicker devices showing lowest reflectivity in the range 500-600 nm, consistent with their absorption spectra. It is notable that the device reflectivity is < 0.2 even for spectral regions where the photoactive layer does not absorb (e.g., > 650 nm), indicative of the impact of parasitic absorption or light scattering from the device stack. It is evident that altering the Cl₆-SubPc thickness from 30 to 50 nm does not result in a significant change in reflectance around Cl₆-SubPc absorbance peak (600 nm), indicating that any interference effects, if present, will be similar for these two devices. The thinnest device PHJ 10/10 nm exhibits slightly higher reflectivity, consistent with its lower absorption, however even in this device, interference effects can be neglected as the photoactive layer thickness (20 nm) is much less than the length scale of expected interference effects^{1,2}. Overall, it can be concluded that the reflectivity of our measured device stacks is too low for interference effects to significantly

impact results and conclusions we report in our study. For example, it is clear that such reflectance / interference effects will be too small to be the primary origin of the strong trend in device EQE with photoactive layer thickness (**Fig. R1b**). We have now added **Fig. R1a** into our supplementary information, and revised the text on page 7 line 157 to refer to this (highlighted in yellow):

*‘The thicker layer will result in significantly higher light absorption. Whilst interference effects may also result in enhanced light absorption, device reflectance data (**Supplementary Fig. 1**) indicate such effects will only be minor and insufficient to explain the observed trend. We note that detailed analyses of such interference effects are beyond the scope of this paper.’*

We also agree that the PL intensity of our OPD devices will be affected by these minor reflection / interference effects. For example, the higher PL intensity of the 10/50 nm PHJ OPD relative to the 10/30 nm OPD (**Fig. 3b & 3c**) in main manuscript may in part result from the higher reflectivity of the 10/50 nm PHJ OPD shown in **Fig. R1a**. This is one reason why we don't attempt to compare the PL intensities of device with different photoactive layers / thicknesses. We have already added a sentence to our manuscript (page 9, line 204) in our previous revision stating this:

‘We note the PL intensities of these devices are likely to be impacted by interference (and possible quenching) effects resulting from the presence of the Al top contact, as such we focus here only on the bias dependence of this device PL.’

We do draw a conclusion from the relative PL intensities of our photoactive films. These films can be expected to show even lower reflectivity due to lack of a reflective metal back contact. The main result we focus on from our film PL data is that the thicker bilayers show higher Cl₆-SubPc PL intensity than a neat Cl₆-SubPc film, in contrast the PL quenching expected for exciton separation at the Cl₆-SubPc / MPTA junction. It is apparent that the trends in Cl₆-SubPc PL we observe in these films are far larger than the changes in reflectivity observed even for our devices. As such, while reflectivity will indeed result in small effects on the PL intensities, they do not impact on the conclusion we draw from our film PL data of an absence of Cl₆-SubPc PL quenching for the 10/30 and 10/50 nm Cl₆-SubPc / MPTA junctions. This allows us to rule out exciton separation at the Cl₆-SubPc / MPTA junction as the primary origin of the high EQE's we observe for the device fabricated with these thicker layers.

Reference

1. Peumans, P., Yakimov, A. & Forrest, S. R. Small molecular weight organic thin-film photodetectors and solar cells. *J Appl Phys* **93**, 3693–3723 (2003).
2. Pettersson, L. A. A. *et al.* Modeling photocurrent action spectra of photovoltaic devices based on organic thin films. *J Appl Phys* **86**, 487–496 (1999).

REVIEWERS' COMMENTS

Reviewer #3 (Remarks to the Author):

The authors investigated the impact of optical interference on the obtained EQE spectrum. I appreciate these additional measurements a lot, all pointing out to a limited effect of optical interference, and hence, strengthening the conclusion of the authors.

The article can be accepted.